# Rewriting the Fate of Cancer: Epigenetic and Epitranscriptomic Regulators in the Metastatic Cascade

**DOI:** 10.3390/biom15111573

**Published:** 2025-11-10

**Authors:** Toshifumi Hara, Murugan Subramanian

**Affiliations:** 1Laboratory of Biochemistry, School of Pharmacy, Aichi Gakuin University, Nagoya 464-8650, Japan; 2Department of Microbiology and Immunology, Weill Cornell Medicine, New York, NY 10065, USA; mus2012@qatar-med.cornell.edu; 3Weill Cornell Medicine–Qatar, Education City, Qatar Foundation, Doha 24144, Qatar

**Keywords:** metastasis, epigenetics, epitranscriptome, ncRNA, miRNA, lncRNA, m^6^A, m^5^C, 5mC, 5hmC

## Abstract

Metastasis is the leading cause of cancer-related mortality, representing a highly coordinated, multistep process in which malignant cells gain the ability to invade, survive in the circulation, and establish secondary tumors at distant sites. While genetic mutations initiate oncogenesis, accumulating evidence shows that epigenetic and epitranscriptomic regulators, encompassing DNA methylation, RNA modifications, and noncoding RNAs (ncRNAs), reshape metastatic phenotypes. This review integrates current insights into these mechanisms and their crosstalk, with a primary focus on their methylation modification. Given their plasticity and potential reversibility, these regulators are attractive targets for therapeutic intervention. Defining the dynamic interplay between DNA and RNA modifications and ncRNAs provides a coherent framework for controlling metastasis and guides the development of precision epigenetic strategies and biomarkers. Future research that integrates multi-omics approaches and spatial transcriptomics will be essential for revealing the epigenetic and epitranscriptomic layers of the metastatic landscape.

## 1. Introduction

Metastasis is the spread of cancer cells from a primary tumor to distant tissues and is the leading cause of cancer-related mortality [1,2]. It unfolds as a multistep invasion–metastasis cascade comprising local invasion of surrounding tissue, intravasation into the vasculature or lymphatics, survival in circulation, extravasation at distant sites, seeding of micrometastases, and outgrowth to form macroscopic secondary tumors [1,3,4,5]. Each step requires transient phenotypic plasticity (for example, epithelial–mesenchymal transition, EMT) and adaptation to diverse microenvironments. As a result, metastasis is biologically complex and inefficient: only a minority of disseminated cells successfully colonize new niches [4,5]. Nevertheless, once established, metastatic disease is frequently incurable, underscoring the urgency of defining mechanisms that could be targeted therapeutically [1,3].

Beyond the well-characterized genetic alterations that initiate and drive tumor progression, converging evidence identifies epigenetic mechanisms as key regulators of metastasis [6]. Epigenetics refers to heritable changes in gene expression that occur without alterations of the DNA sequence. These includes covalent DNA modifications (for examples, cytosine methylation), RNA modifications, histone modifications and chromatin remodeling, and regulation by noncoding RNAs (ncRNAs) such as microRNAs (miRNAs), long noncoding RNAs (lncRNAs), and circular RNAs (circRNAs) [6,7,8,9]. These epigenetic marks are dynamic and reversible, allowing for rapid cellular adaptation in various environments. In many cancers, global epigenomic reprogramming accompanies the acquisition of metastatic competence; comparative analyses of primary and metastatic lesions often reveal aberrant DNA methylation and dysregulated ncRNAs networks [6,8,9,10]. Such alterations confer pro-metastatic traits, including loss of adhesion, increased motility, stem-like features, and immune evasion, without requiring additional genetic change [1,3,10].

This review summarizes current knowledge on three major classes of epigenetic and epitranscriptomic regulators that shape metastasis in solid tumors: DNA methylations, RNA modifications, and ncRNAs (Figure 1).

We describe how these mechanisms reprogram gene expression and cellular state to drive metastatic progression, using examples from breast, lung, colorectal, prostate, and other cancers. We also discuss translational implications, including the development of epigenetic marks and ncRNAs as biomarkers of metastasis and as therapeutic targets. By integrating recent advances, we provide an up-to-date perspective on how epigenetic plasticity fuels metastatic evolution and how it may be leveraged to improve clinical outcomes.

## 2. DNA Methylation in Metastasis

DNA methylation is the addition of a methyl group to the 5-carbon of cytosine (5-methylcytosine, 5mC) within CpG dinucleotides. It is a foundational epigenetic mechanism that is frequently altered in cancer [11]. Aberrant DNA methylation patterns, including locus-specific hypermethylation (often in gene promoter CpG islands) and global hypomethylation, are hallmarks of cancer and contribute directly to metastasis [12]. Hypermethylation at gene promoters typically induces chromatin condensation and gene silencing; in cancer, this often inactivates genes that restrain tumor progression [13] (Table 1).

For example, the cell–cell adhesion molecule E-cadherin (encoded by *CDH1*) is a well-known tumor suppressor whose expression is frequently lost in metastatic carcinomas due to promoter CpG island hypermethylation [14,16,27]. Silencing of E-cadherin through DNA methylation diminishes intercellular adhesion, facilitating EMT and invasion, thereby promoting metastasis and correlating with a poor prognosis in multiple cancers [14,28]. Many other metastasis-suppressor genes such as *TIMP3*, *BRMS1*, *SEMA3E*, are likewise reported to undergo promoter hypermethylation during cancer progression, contributing to a more invasive phenotype [20,22]. Accordingly, increased expression or activity of DNA methyltransferases (DNMTs), the enzymes that catalyze 5mC formation, is often observed in advanced tumors [29]. Clinical studies have shown that the overexpression of DNMT1, DNMT3A, or DNMT3B in primary tumors is associated with increased risk of metastasis and poor outcomes [15,26]. In prostate cancer, upregulation of DNMT1 and DNMT3 is associated with a higher incidence of lymph node metastases [17]. In hepatocellular carcinoma, the upregulation of DNMT1 is associated with the repression of E-cadherin and the development of larger, more invasive tumors [18]. These findings suggest that abnormal DNA methylation is not just a consequence of tumorigenesis but actively contributes to the metastatic potential of tumors.

Conversely, hypomethylation of genomic DNA, such as the demethylation of normally methylated repetitive elements and heterochromatin regions, is another common alteration seen in metastatic cancers [12]. A widespread reduction in 5mC can result in chromosomal instability and the reactivation of transposable elements, which promotes genetic diversity and adaptability in tumor cells [30]. Hypomethylation at specific gene loci can also cause aberrant overexpression of pro-metastatic genes such as *PRAME* and *S100A4*, which are kept repressed in normal cells [23,31,32]. Thus, the shifting DNA methylation landscape in metastasis involves both targeted hypermethylation (shutting down inhibitors of metastasis) and diffuse hypomethylation (facilitating the permissive expression of genes that promote metastasis and instability). This dual remodeling of the methylome drives acquisition of metastatic traits, a paradoxical state of promoter hypermethylation alongside global hypomethylation. DNA methylation changes in metastasis can also affect noncoding genomic regions that control gene expression. For instance, enhancer or insulator methylation changes may activate pro-metastatic pathways or turn off genomic imprinting/tumor suppressive circuits [33]. In fact, hypermethylation can silence certain miRNA genes that normally suppress metastasis. The miR-34 family of tumor-suppressor miRNAs is frequently downregulated in metastatic disease; this downregulation has been attributed, in part, to CpG methylation of the *MIR34A/B/C* promoters in cancers such as colon, breast, and lung [34]. Loss of miR-34 permits increased expression of targets such as c-MET and SNAIL, which drive invasion and metastasis [35]. More broadly, a genome-wide study identified a DNA methylation signature that involves the silencing of multiple metastasis-suppressing miRNAs in metastatic tumors [36]. These findings reinforce that DNA methylation interacts with other epigenetic regulators, such as miRNAs, to orchestrate metastatic behavior.

Dynamic regulation of 5mC by DNA demethylases can also influence metastasis. The Ten-eleven translocation (TET) family of enzymes plays a crucial role in the oxidation of 5mC to 5-hydroxymethylcytosine (5hmC), a key step in the process of DNA demethylation [37]. A decrease in 5hmC is often found in advanced cancers and metastases, indicating reduced TET activity [38]. In some cases, TET-mediated demethylation can inhibit metastasis by reactivating genes that have been suppressed [39]. In addition, aberrant demethylation might activate genes that drive metastasis. An example is miR-29, which directly targets TET transcripts. In hepatocellular carcinoma, miR-29a induces the loss of 5hmC and promotes metastasis via a TET-SOCS1-MMP9 axis [19]. Here, miR-29a downregulates TET expression, leading to hypermethylation and silencing of the *SOCS1* gene (a negative regulator of STAT3/MMP9 signaling), which in turn permits upregulation of MMP9 and aggressive, invasive behavior. This example highlights a convergence of epigenetic mechanisms, including DNA methylation and miRNAs, in driving metastasis.

On the other hand, excessive TET activity may facilitate metastasis under specific contexts, as demethylation can increase the expression of pro-metastatic factors. The specific outcome, however, depends on the targeted genes and the tissue environment. Translationally, insights into the role of DNA methylation in metastasis have spurred efforts to exploit these marks as biomarkers and therapeutic targets. Hypermethylated gene promoters can serve as blood-based DNA methylation biomarkers signaling occult metastatic disease or recurrence (for example, *CDH1* or *RARB* methylation in circulating tumor DNA) [40,41,42]. DNA methylation patterns in primary tumors may also predict metastatic proclivity; for example, a “metastatic methylation signature” has been proposed for specific cancers [43]. In terms of therapy, hypomethylating agents (DNA methyltransferase inhibitors, such as 5-azacytidine and decitabine) can reactivate metastasis-suppressor genes that are silenced by methylation [44]. These drugs are already used in hematologic malignancies and are being investigated in solid tumors [45,46]; there is interest in whether they could reverse EMT or prevent metastatic outgrowth by restoring expression of adhesion molecules and other suppressors [47,48]. However, current systemic DNMT inhibitors induce broad CpG hypomethylation and can inadvertently activate oncogenes. By contrast, precise targeting of locus- or cell-restricted epigenome editing can modulate methylation at a defined regulatory element while sparing the rest of the genome. At present, CRISPR/dCas9-TET1 to demethylate a silenced metastasis suppressor or dCas9-DNMT3A to methylate a single enhancer that drives invasion; these approaches are compelling at the bench but remain preclinical [49,50]. Another approach is to target upstream regulators of aberrant methylation, such as inhibiting overexpressed DNMTs or modulating TET activity, to rebalance the methylome. Overall, DNA methylation is a double-edged sword in metastasis: it can serve as an early-warning biomarker of aggressive behavior and as a potentially malleable point of intervention to restrain metastatic progression.

## 3. RNA Epigenetic Modifications in Metastasis

DNA methylation influences the encoding of genomic information, but a comparable regulatory framework at the RNA level has also recently emerged as the epitranscriptome. The epitranscriptome comprises RNA modifications applied to RNA nucleotides after transcription, which can significantly impact RNA stability, processing [51], and translation. Thus, gene functions can also be modified without changing of the DNA sequence by the epitranscriptome. More than 170 distinct RNA modifications, including pseudouridine (Ψ), m^1^A, ac^4^C, m^7^G, and A-to-I editing, have been identified in cells. Among these, N^6^-methyladenosine (m^6^A) and 5-methylcytidine (m^5^C) are two of the most intensively studied marks in cancer biology [7,52]. This review focuses on m^6^A and m^5^C among the diverse RNA modifications. Dysregulation of these RNA modifications, particularly m^6^A and m^5^C, has been identified as a crucial contributor to the development of oncogenesis and metastasis [53,54]. In this section, we focus on how m^6^A and m^5^C RNA modifications, along with their writer/eraser/reader proteins, drive metastatic progression (Table 2).

### 3.1. m^6^A RNA Methylation and Metastasis

N^6^-methyladenosine, m^6^A, is the most abundant internal modification in eukaryotic messenger RNAs and ncRNAs. The dynamic m^6^A methylome on RNA transcripts has been comprehensively revealed by m^6^A-specific sequencing techniques [55]. This mark, installed at the N^6^ position of adenine, affects RNA metabolism at multiple levels, including splicing, nuclear export, decay, and translation, depending on the context and the “reader” proteins that recognize a specific m^6^A [56]. The m^6^A landscape is dynamically regulated by “writer” enzymes (the METTL3/METTL14/WTAP methyltransferase complex and related factors) that add the mark, and “eraser” enzymes (FTO and ALKBH5 demethylases) that remove it [57]. Distinct m^6^A-binding reader proteins, such as the YTH domain family and IGF2BP family, then translate the presence of m^6^A into functional outcomes, such as altered mRNA stability or translation efficiency [58]. Aberrant regulation of m^6^A has been implicated in many aspects of tumor biology, including metastasis. Global m^6^A levels in mRNA often change during cancer progression, although findings are sometimes context-dependent [59]. Many aggressive tumors show increased m^6^A modification on transcripts, partly due to overexpression of m^6^A writers. For example, METTL3, the catalytic core component of the m^6^A methyltransferase complex, is frequently upregulated in advanced cancers such as lung adenocarcinoma, hepatocellular carcinoma, pancreatic cancer, and colorectal carcinoma [60,61,62,63]. High *METTL3* levels have been correlated with higher metastatic capacity and poor survival (Table 3).

In colorectal cancer, *METTL3* expression is elevated in metastatic tumors, and *METTL3* drives cell migration, invasion, and metastasis [63]. Mechanistically, *METTL3* can promote metastasis by enhancing the maturation or translation of pro-metastatic transcripts. In a colorectal study, METTL3-mediated m^6^A on pri-miR-1246 was shown to increase processing of this miRNA, which in turn downregulated SPRED2 (a negative regulator of MAPK signaling), thereby unleashing MAPK activity and metastasis [64]. Similarly, in oral squamous cell carcinoma and pancreatic cancer, METTL3-catalyzed m^6^A installation on specific mRNAs such as *BMI1* or E2F family transcripts leads to their increased translation and confers invasive, stem-like traits to tumor cells [76,79]. These studies illustrate that *METTL3* has pro-metastatic functions by reprogramming the epitranscriptome to favor cancer cell motility and survival.

While m^6^A is often pro-metastatic, its role can vary and may be metastasis-suppressive in specific contexts (Figure 2).

In triple-negative breast cancer (TNBC), *METTL3* expression was relatively low; restoring METTL3 increased m^6^A on the *COL3A1* mRNA, which led to reduced COL3A1 expression and impaired invasion [70]. In this case, reduced METTL3 allowed *COL3A1* (a collagen gene involved in extracellular matrix remodeling) to be aberrantly upregulated, consequently enhancing metastatic spread. Thus, *METTL3* can have tumor-suppressive effects in certain cellular contexts by modulating metastasis-relevant genes. Likewise, the other m^6^A writer enzyme METTL14 exhibits tumor-type specific roles. Although METTL14 promotes metastasis by increasing m^6^A on *PERP* mRNA in pancreatic cancer metastasis, it often acts as a metastasis suppressor in colorectal and hepatocellular carcinoma, where loss of METTL14 leads to EMT and stemness through reduced m^6^A abundance on key transcripts [71,77,82]. Thus, METTL14 exhibits tumor-type-dependent roles in metastasis, underscoring that m^6^A machinery must be interpreted within disease context and cellular wiring. For example, METTL14-mediated m^6^A on the *SOX4* transcript in colorectal cancer inhibits *SOX4* expression and metastasis; low METTL14 permits SOX4 protein upregulation and is associated with metastatic relapse [66]. These divergent observations underscore that the impact of m^6^A modification on metastasis depends on which RNAs are targeted and the cellular environment [59].

In addition to writers, m^6^A erasers and readers contribute to metastasis. The demethylase *FTO* is overexpressed in cancers such as breast and melanoma and can promote metastasis by erasing m^6^A marks on mRNAs that encode pro-metastatic proteins, thereby stabilizing those transcripts [67,80]. Conversely, the other eraser *ALKBH5* has been reported to suppress metastasis in some contexts, such as breast cancer, by demethylating and thereby stabilizing transcripts of tumor-suppressive genes or by interfering with pre-mRNA splicing of EMT drivers [69]. m^6^A reader proteins of the YTHDF family influence metastasis by binding m^6^A-marked transcripts and altering their fate [68,72]. YTHDF1, which promotes translation, is often upregulated in metastatic cancers and linked to worse outcomes [72,83]. YTHDF1 enhances protein abundance encoded from pro-metastatic mRNAs and is being studied as a potential prognostic marker in various cancers [72,84]. By contrast, YTHDF2 accelerates mRNA decay and may inhibit metastasis by degrading mRNAs that encode metastasis-promoting factors, although it can also destabilize mRNAs of metastasis suppressors [81,85].

Another class of m^6^A readers, the IGF2BP1/2/3 proteins, bind m^6^A-modified transcripts and greatly stabilize them. *IGF2BP3*, in particular, is highly expressed in aggressive tumors and was shown to enhance the stability of m^6^A-modified mRNAs such as *MYC* or *SLUG*, thereby promoting invasion and metastatic colonization [58,73,74]. High *IGF2BP3* level expression correlate with poor survival of cancer patients, like colorectal and pancreatic cancer, consistent with a pro-metastatic function [65,78].

Taken together, these findings paint a picture in which m^6^A dysregulation is a pervasive feature of metastatic progression. By rewiring post-transcriptional gene expression programs, changes in m^6^A deposition or recognition can tip the balance at epithelial–mesenchymal states, or between dormancy and proliferation of disseminated cells. Importantly, components of the m^6^A machinery are emerging as drug targets for cancer therapeutics. Several small-molecule inhibitors of *METTL3* have been developed, and their effects on solid tumor metastasis are under investigation. Inhibiting *METTL3* or an m^6^A reader could, in principle, reduce metastatic seeding in cancers where m^6^A drives metastasis. Conversely, restoring the levels of m^6^A or inhibiting erasers would be beneficial in patients with contexts where m^6^A has tumor-suppressive roles. The therapeutic window and safety of these strategies are active areas of cancer research. Nevertheless, targeting the “epitranscriptome” holds promise because, like other epigenetic alterations, RNA modifications are reversible and do not require that underlying genetic mutations be fixed. Precision therapies, such as modified antisense oligonucleotides targeting m^6^A on specific transcripts, may significantly reduce a tumor’s metastatic potential with high specificity. In summary, m^6^A dysregulation reshapes RNA fate to tip the balance between dormancy and invasion, underscoring its therapeutic potential.

### 3.2. m^5^C RNA Methylation and Metastasis

5-methylcytidine on RNA (m^5^C), analogous to DNA 5-methylcytosine, is an epitranscriptomic mark increasingly recognized for its role in cancer biology. m^5^C was long known to occur in ncRNAs, including rRNA and tRNAs, but recent high-resolution techniques in high-throughput mapping have shown that m^5^C also exists on mRNAs and lncRNAs [86,87,88]. Although m^5^C is a comparatively rare modification, it can exert significant influence on RNA fate, affecting RNA stability, translation, localization, and forming binding sites for specific proteins [89]. The enzymatic writers of RNA m^5^C include the NSUN family (*NSUN1* through *NSUN7*) and *DNMT2*, which methylate cytosine residues in different RNA contexts [90]. The erasers of m^5^C remain unclear, but the TET enzymes and ALKBH1 have been suggested to generate 5-hydroxymethylcytidine (hm^5^C) or otherwise demethylate m^5^C [91,92]. Importantly, several reader proteins recognize m^5^C marks, notably *ALYREF* (an mRNA export adaptor) and Y-box binding protein 1 (*YBX1*) [93,94]. These readers bind methylated cytosines and commonly promote export or stabilization of the modified transcripts. Emerging evidence indicates that dysregulation of m^5^C and its associated proteins contributes to tumor progression and metastasis [95]. *NSUN2*, a cytosine methyltransferase, is a well-studied m^5^C writer in cancer. NSUN2 targets a range of RNAs, including tRNAs, mRNAs, and ncRNAs, and is frequently overexpressed in aggressive tumors [93,94,96]. In addition, NSUN2 often promotes metastasis by methylating specific mRNAs to enhance their stability and translation through m^5^C reader proteins (Table 4).

In bladder cancer, NSUN2 methylates the 3′ untranslated region (3′-UTR) of the *HDGF* (hepatoma-derived growth factor) mRNA, installing m^5^C marks that are bound by YBX1 [94]. YBX1 binding protects the transcript from degradation, leading to higher HDGF protein levels and increased tumor cell growth, invasion, and metastasis [94,97]. NSUN2 and YBX1 thus act in concert as writer and reader to post-transcriptionally upregulate pro-metastatic factors like HDGF. Similarly, in breast cancer and melanoma, *NSUN2* overexpression correlates with advanced stage and metastasis; NSUN2 can methylate mRNAs or even tRNA-derived fragments that regulate cell motility and cytoskeletal dynamics, thereby facilitating metastatic spread [107,108]. The oncogenic function of NSUN2 is underscored by studies showing that *NSUN2* knockout or knockdown impairs migration and metastasis in models of esophageal squamous carcinoma, glioma, and others [98,101,109]. In esophageal cancer, NSUN2-mediated m^5^C on a specific lncRNA was found to promote that lncRNA’s association with a chromatin modifier (BPTF), increasing levels of matrix metalloproteinases (MMPs), which leads to degradation of the extracellular matrix, promoting invasion [99,100]. This highlights that RNA m^5^C can interface with chromatin regulation indirectly through lncRNAs. Other NSUN family members also contribute to metastasis. *NSUN5* and *NSUN6* have been linked to metastasis in certain cancers. For example, in breast cancer, *NSUN6* expression is elevated in bone metastases, and NSUN6-mediated m^5^C on specific transcripts is associated with activation of YAP signaling, promoting osteotropic metastasis [110,111,112]. *NSUN5*, on the other hand, has been implicated in metastasis of head and neck squamous cell carcinoma and gliomas, although its precise targets in those contexts are still being elucidated [113,114].

Beyond the NSUN enzymes that target mRNA, the DNMT2 enzyme primarily methylates tRNA but may indirectly influence metastasis by affecting protein synthesis and cell stress responses [115,116]. The overall picture is that m^5^C “writer” enzymes commonly act as metastasis promoters by reprogramming the transcriptome of cancer cells at the RNA level. Crucially, the reader proteins of m^5^C mediate the downstream effects of these methylation events. ALYREF, a nuclear RNA-binding protein that shuttles mRNAs to the cytoplasm, binds preferentially to m^5^C-modified transcripts and enhances their export from the nucleus [93]. Overexpression of *ALYREF* in cancers has been associated with increased translation of m^5^C-marked oncogenic mRNAs. ALYREF was also found to bind and export transcripts modified with m^5^C of *MYC*, *YAP*, and *PKM2*, leading to their increased expression and contributing to tumor progression and resistance to therapy [93,117,118]. Meanwhile, *YBX1*, a cytoplasmic mRNA-binding protein, is another critical m^5^C reader in metastasis. YBX1 has high affinity for m^5^C sites in the 3′-UTRs or coding sequences of target mRNAs, and upon binding, YBX1 can recruit stabilizing factors like PABP (poly(A)-binding protein) to protect the transcript [94,119,120]. YBX1 also often enhances translation of its target mRNAs. Of note, YBX1 recognizes m^5^C sites on *FOXC2*, *HDGF*, and *AR* encoding pro-metastatic proteins, resulting in the enhancement of translation efficiencies and functions [94,102,104]. FOXC2 is a transcription factor that drives EMT and metastasis. As HDGF facilitates tumor angiogenesis and growth, YBX1’s stabilization of these mRNAs directly contributes to the metastatic phenotype. In cholangiocarcinoma (bile duct cancer), YBX1 binding to m^5^C-modified *NKILA* lncRNA was reported to stabilize that lncRNA and promote cancer cell migration [121]. Intriguingly, *NKILA* is known to interact with NF-κB signaling, suggesting an epigenetic feed-forward loop where m^5^C, lncRNAs, and inflammatory signaling intersect in metastasis [121,122].

Overall, the aberrant upregulation of m^5^C readers like YBX1 in cancers creates a scenario in which any mRNA bearing m^5^C might be aberrantly stabilized, tilting the proteome toward a metastatic state. On the other hand, disruption of the m^5^C machinery can impair metastasis. Loss-of-function of NSUN2 in cancer cells leads to reduced methylation of many RNAs, often resulting in decreased stability of pro-migratory transcripts and a subsequent decline in invasion [94,105]. Depletion of YBX1 or ALYREF has similarly been shown to sensitize m^5^C-marked transcripts to degradation and to reduce metastasis in experimental models [106,123]. These findings suggest that targeting m^5^C writer or reader proteins could be a viable therapeutic strategy. Indeed, NSUN2 is being explored as a potential drug target or prognostic marker. The levels of *NSUN2* expression in tumors have prognostic value in certain cancers, such as high NSUN2 portends poor outcome and greater metastatic risk [103,105,124]. YBX1 has also been investigated as a therapeutic target in aggressive cancers, although being a DNA/RNA-binding protein makes it challenging to target directly. An alternative approach might be to disrupt the interaction between m^5^C marks and their readers. Small molecules or peptides that mask m^5^C sites could prevent readers like YBX1 from binding, thereby destabilizing those transcripts [125]. Such approaches remain speculative but exemplify the new avenues of intervention that epitranscriptomic research is just beginning to open. In summary, RNA m^5^C modification is now recognized as an important regulator of metastasis. Through m^5^C modification on coding and ncRNAs, cancer cells can post-transcriptionally upregulate entire cohorts of genes that endow metastatic capabilities, including invasion, anoikis resistance, and colonization. The NSUN methyltransferases and YBX1/ALYREF reader proteins form an axis that frequently drives metastasis in diverse tumors by synergistically methylating and stabilizing metastasis-related RNAs. RNA modifications can quickly alter protein expression without new transcription. This ability is particularly useful for cancer cells during metastasis. As detection methods for m^5^C improve, such as high-resolution m^5^C RNA sequencing, we are likely to discover specific “m^5^C-regulated RNA networks” that control metastatic colonization and perhaps metastatic latency. Investigating components of the m^5^C regulatory network and utilizing m^5^C markers as potential biomarkers, such as altered tRNA methylation patterns in circulating vesicles, may provide new approaches to combat metastatic cancer. Overall, aberrant m^5^C regulation establishes a post-transcriptional program that supports invasion and colonization, highlighting the NSUN–YBX1/ALYREF axis as a therapeutic opportunity.

## 4. Noncoding RNAs in Metastasis

ncRNAs are RNA molecules that are not translated into proteins. In this review, we explicitly separate housekeeping ncRNAs (rRNA, tRNA, snRNA, snoRNA) from regulatory ncRNAs (miRNAs, lncRNAs, circRNAs). Regulatory ncRNAs have emerged as central players in post-transcriptional and epigenetic regulation of gene expression and are particularly implicated in cancer phenotypes such as metastasis. In various human cancers, multiple pathways by which ncRNAs regulate metastasis through RNA modifications and epigenetic mechanisms have been elucidated. miRNAs suppress or promote metastasis by altering histone modifications and DNA methylation states through regulating the expression levels of target epigenetic factors. Meanwhile, lncRNAs and circRNAs can comprehensively regulate metastasis-related gene expression by recruiting epigenetic factors as scaffold molecules or by acquiring functions through RNA modifications applied to themselves. These findings suggest that ncRNAs occupy a central position within the epigenome and epitranscriptome of the regulatory network governing the complex process of metastasis. Here, we focus on representative ncRNAs, miRNA, and lncRNA, to discuss the relationship between cancer metastasis and RNA modifications and epigenetics.

### 4.1. microRNAs and Metastasis

miRNAs are ~20–24 nucleotide small RNAs that typically bind to complementary sequences in the 3′-UTRs of target mRNAs, inducing mRNA degradation or translational repression [126,127]. A single miRNA can regulate dozens of genes mainly by the seed-region of miRNA, and conversely, a given mRNA may be targeted by multiple miRNAs, making them potent modulators of gene networks [128]. In cancer, numerous miRNAs are aberrantly expressed, functioning either as oncogenic miRNAs or tumor-suppressor miRNAs [129] (Table 5).

#### 4.1.1. RNA Modification of miRNAs

RNA modifications are known to impact miRNA functions in various ways, affecting cancer phenotypes. N^6^-methyladenosine (m^6^A) marks on primary miRNAs (pri-miRNAs) modulate the activity of Microprocessor, which consists of Drosha and DGCR8 [130]. The nuclear RNA-binding protein HNRNPA2B1 acts as an m^6^A “reader” on a subset of pri-miRNAs, physically associates with DGCR8, and thereby promotes pri-miRNA processing [131]. Thus, loss of HNRNPA2B1 phenocopies METTL3 depletion for those substrates. Therefore, the METTL3/METTL14–m^6^A–HNRNPA2B1–DGCR8 axis affects the abundance of mature miRNA. In hepatocellular carcinoma (HCC), METTL14 binds DGCR8 and facilitates m^6^A-dependent processing of pri-miR-126 [71]. Reduced METTL14 expression represses the mature miR-126 level, enhancing migratory and invasive activity. In orthotopic and tail-vein models, downregulation of METTL14 increased metastatic burden, while restoring METTL14 or miR-126 suppressed dissemination, linking an m^6^A-programmed miRNA defect to metastasis in vivo [71,132]. Additionally, m^6^A-sensitive pri-miRNAs in HCC reinforce the notion that m^6^A-guided processing is a metastasis checkpoint [133,134].

METTL1 installs 7-methylguanosine (m^7^G) into specific pri-/pre-miRNAs. In the let-7 family, m^7^G disrupts a form of G-quadruplex, enhancing the stability of let-7 maturation and reduction in migration activity [135]. By a mass spectrometry approach, m^7^G modification was identified in a single guanosine in let-7e-5p, and genetic or catalytic inactivation of METTL1 reduces mature let-7 form and increases migratory activity [135]. While these data are limited to migration/invasion rather than overt metastasis, they pin a defined chemical mark on a defined miRNA to a pro-metastatic phenotype in human cancer cells.

At the 3′ ends of miRNA, terminal uridylyl transferases TUT4/ZCCHC11 and TUT7/ZCCHC6, recruited by LIN28A/B to the conserved GGAG element in pre-let-7, catalyze oligo-uridylation [136]. The uridylation on pre-let-7 prevents Dicer cleavage and suppresses the maturation of let-7. In this process, LIN28 switches TUT4/7 from mono- to oligo-uridylation to enforce this block [137]. In contrast, in somatic cells lacking LIN28, mono-uridylation facilitates the processing of group II pre-let-7 [138]. Functionally, the LIN28–TUT4/7 uridylation gate sits directly on migratory and invasive activity through the let-7 network.

Adenosine-to-inosine (A-to-I) editing, changing a specific nucleotide base within the seed region of miRNAs, affects the recognition of target genes by miRNA [139]. In melanoma, a decrease in the editing of miR-455-5p changes its role from a suppressor of growth and metastasis (the edited form) to a regulator that promotes metastasis (the unedited form) [140]. Either restoring ADAR1 or expressing the edited form of miR-455-5p reduces tumor growth and metastases in the lung metastasis model [141]. Conversely, the unedited isoform of miR-455-5p suppresses CPEB1 and encourages tumor dissemination [140,141]. Additionally, attenuated editing of miR-376a* enhances migration and invasion in vitro and increases aggressiveness in orthotopic glioma models [142]. Although brain tumors do not conventionally “metastasize” outside the CNS, this study demonstrates that editing state dictates invasive behavior in vivo. These experiments provide a direct causal chain from miRNA editing status to metastatic outcome.

#### 4.1.2. Epigenetics of miRNAs

miRNAs influence cancer cell functions in various ways, both directly and indirectly, through epigenetic mechanisms. Loss of miR-101 relieves repression of EZH2, the catalytic subunit of PRC2, expanding H3K27me3 at anti-metastatic loci and shifting chromatin toward a pro-invasive program [143]. In breast cancer patients, progression to advanced and metastatic disease is accompanied by decreased miR-101 and reciprocal EZH2 upregulation, and functional studies show that restoring miR-101 restrains invasion by lowering EZH2 and its repressive mark [143,144]. Collectively, these data position the miR-101-PRC2/EZH2 axis as a core epigenetic route to metastatic competence in various tumors.

In gastric cancer, miR-29b/c and DNMT3A form a methylation circuit that converges on CDH1. Loss of miR-29b/c with gain of DNMT3A drives *CDH1* promoter hypermethylation, represses E-cadherin, and increases migration/invasion, an epigenetic route to EMT and metastatic competence [21]. Thus, direct targeting of DNMT3A by miR-29b/c is associated with methylation-dependent silencing of miR-29b/c itself.

miR-148a directly suppresses DNMT1, leading to promoter demethylation and re-expression of metastasis-suppressor genes in pancreatic cancers [24,25]. Further, restoring miR-148a (or inhibiting DNMT1) reduces migration and invasion, indicating that the DNMT1–miR-148a axis functions as an epigenetic brake on metastatic traits [25]. In clinical samples, reduced miR-148a expression correlated with increased expression of DNMT1, consistent with pathway disruption in aggressive disease [24,25].

As in breast cancer, miR-101 loss in prostate tumors unleashes EZH2 and H3K27me3-mediated silencing across metastasis-suppressive programs in prostate cancer [143,145]. Genomic analyses reveal frequent deletions of miR-101 loci in metastatic samples. In addition, functional studies indicate that restoring miR-101 reduces EZH2 expression and invasive behavior, making this miR-101–PRC2 axis one of the most precise miRNA-driven epigenetic mechanisms linked to metastasis [143,146,147].

Taken together, the functional regulation of cancer metastasis through complex feedback loops and networks involving miRNAs, epigenetics, and RNA modifications is only beginning to be elucidated.

### 4.2. Long Noncoding RNAs and Metastasis

lncRNAs are a broad class of ncRNAs of over 200 nucleotides in length that do not encode proteins. lncRNAs are involved in gene regulation through diverse mechanisms. They can act as scaffolds for protein complexes, guides to recruit chromatin modifiers to specific genomic loci, decoys that sequester proteins or miRNAs, or even as precursors for small RNAs. The human genome encodes thousands of lncRNAs, and aberrant lncRNA expression is a common feature of cancer. Many lncRNAs have been identified as key drivers or suppressors of metastasis, often functioning in a cell-type-specific manner (Table 6).

#### 4.2.1. RNA Modification of lncRNAs

*HOTAIR* is a well-established lncRNA that rewires chromatin in favor of metastatic activity in multiple cancer types [148,149,167]. *HOTAIR* carries a conserved m^6^A site (A783) that is required for its chromatin repression program [164,168]. Mutating this single site weakens *HOTAIR*’s interaction with the nuclear m^6^A reader YTHDC1, blunts H3K27me-linked silencing, and reduces malignant growth and invasion of breast cancer cells [148,164,168]. In patient datasets, tumors with high *HOTAIR* and intact A783-dependent signaling show stronger Polycomb target repression and worse outcomes, consistent with an epitranscriptomic “licensing” of HOTAIR function [148,169].

One of the classical lncRNAs, *H19*, is a direct substrate of NSUN2 in hepatocellular carcinoma. m^5^C on *H19* increases transcript stability and promotes the recruitment of the oncoprotein G3BP1, supporting proliferation and tumor progression; higher *H19* m^5^C and expression levels are associated with poor differentiation in patients with liver cancer [96,98].

METTL14, in part, promotes m^6^A-dependent decay of oncogenic *XIST* transcripts through YTHDF2. Dampening *XIST* through this pathway reduces proliferation and metastatic potential in models and correlates with improved prognosis of the patients with colorectal cancer [82,165].

The lncRNA, RNA Component of Mitochondrial RNA Processing Endoribonuclease (*RMRP*), carries abundant m^6^A, which increases its stability. *RMRP* recruits DNA methyltransferases to the *SCARA5* promoter, induces promoter methylation, and suppresses SCARA5 expression. Silencing RMRP reduces proliferation, migration, and invasion of bladder cancer [170]. *LINC01833* is m^6^A-modified by METTL3 and engages hnRNPA2B1 [171]. This module promotes growth and invasive traits of NSCLC, which are associated with adverse clinicopathologic features [172]. *PCAT6* is m^6^A-modified and bound by the reader IGF2BP2, which stabilizes *IGF1R* mRNA. This lncRNA–reader circuit fuels proliferation and bone metastasis of prostate cancer in mouse models and is enriched in high-risk clinical situations [58].

YTHDF3, an m^6^A reader, recognizes m^6^A-modified *GAS5* and accelerates its decay in colorectal cancer [173]. Although the binding of *GAS5* to YAP promotes its phosphorylation and ubiquitylation for degradation, loss of *GAS5* lifts the restraint on YAP signaling, resulting in enhancement of cell proliferation, migration, and further in vivo tumor progression [173,174]. In a clinical study, a low *GAS5* level is inversely correlated with YTHDF3 and YAP protein levels in colorectal tumors [173].

The host gene of the miR-100, *MIR100HG*, binds the m^6^A reader/adapter hnRNPA2B1 to stabilize m^6^A-modified *TCF7L2* mRNA, amplifying Wnt target expression and driving colorectal tumor growth and invasion [131,161,162]. Disruption of METTL3-installed m^6^A or of the *MIR100HG*–hnRNPA2B1 interaction destabilizes *TCF7L2* and curtails malignant behavior [162,163].

The m^6^A eraser ALKBH5 binds and demethylates *NEAT1* transcript in gastric cancer, increasing *NEAT1* activity [154]. Elevated ALKBH5/*NEAT1* enhances invasion and metastasis, in part by modulating *EZH2* expression and downstream EMT programs [155]. Suppression of *NEAT1* reverses these effects in vitro and in mouse models [156]. In addition, *NEAT1* is m^6^A-modified in aggressive prostate cancer [157]. m^6^A enhances *NEAT1*’s nuclear functions that foster chromatin remodeling and transcriptional programs conducive to osteotropism, increasing bone colonization in vivo; genetic interference with NEAT1 or m^6^A recognition suppresses bone metastasis [158]. Furthermore, reduced level of ALKBH5 provides m^6^A accumulation on the metastasis-restraining lncRNA *KCNK15-AS1* in pancreatic ductal adenocarcinoma. Restoration of ALKBH5 expression recovers demethylation and stabilization of KCNK15-AS1, suppressing migration and invasion [159]. *KCNK15-AS1*, in turn, modulates PTEN–AKT signaling and downstream motility circuits [160].

Collectively, the relations between lncRNA and RNA modification hard-wire EMT, motility, survival under stress, and metastatic colonization into the cancer transcriptome.

#### 4.2.2. Epigenetics of lncRNAs

Overexpressed *HOTAIR* binds PRC2 through EZH2 and the LSD1/CoREST complex and retargets them across the genome, increasing H3K27me3 and reducing H3K4 methylation at anti-metastatic loci [148,149,150]. These changes enhance invasion in vitro and promote distant colonization in mouse models, and high *HOTAIR* levels in primary tumors correlate with subsequent metastasis and poorer survival.

*GClnc1* illustrates a chromatin-scaffolding lncRNA in gastric cancer [151]. It assembles WDR5 (to promote H3K4 methylation) and KAT2A/GCN5 (to promote histone acetylation) at specific promoters, establishing modification patterns that support EMT, migration, and invasion; expression rises with tumor size, depth of invasion, and nodal spread [152].

Aggressive prostate cancers often activate *SChLAP1*, which directly antagonizes the SWI/SNF (BAF) chromatin-remodeling complex [175]. Gain- and loss-of-function studies show that *SChLAP1* impairs SWI/SNF occupancy genome-wide, shifting chromatin states toward oncogenic, pro-metastatic transcriptional programs; this remodeling increases invasion and distant spread and is enriched in high-risk clinical disease [153,176].

Across organs, a recurring theme is that lncRNAs reshape the epigenome by recruiting or opposing core chromatin regulators (PRC2, LSD1/CoREST, SWI/SNF) or, in some cases, by exploiting RNA modifications (for example, m^6^A installed by METTL3) to stabilize scaffolds that then alter chromatin-linked signaling. The downstream effect is a durable transcriptional program that supports EMT, invasion, survival in circulation, and colonization at distant sites.

### 4.3. circRNAs and Metastasis

Circular RNAs (circRNAs) arise from back-splicing of pre-mRNAs, yielding covalently closed loops that are typically exon-derived, lack 5′/3′ ends, and are relatively resistant to exonucleases, enabling stable accumulation in cancer cells and extracellular vesicles [177]. Their subcellular localization (nuclear vs. cytoplasmic) and binding partners dictate function.

Epitranscriptomic modification of circular RNAs, particularly m^6^A, regulates biogenesis, nuclear export, subcellular localization, translation, and decay, which together shape post-transcriptional programs relevant to EMT, invasion, survival in circulation, and colonization [178]. m^6^A marks on circRNAs are recognized by nuclear and cytoplasmic readers that direct RNAs toward export, degradation, or cap-independent translation, establishing routing decisions that can influence metastatic behavior in specific contexts [178].

A metastasis-focused exemplar is *circNSUN2* in colorectal cancer, where m^6^A on *circNSUN2* is recognized by YTHDC1 and promotes nuclear export to the cytoplasm [179]. In the cytoplasm, *circNSUN2* scaffolds IGF2BP2 on *HMGA2* mRNA, which stabilizes *HMGA2* and enhances liver metastatic colonization, and loss-of-function for *circNSUN2*, YTHDC1, or IGF2BP2 reduces *HMGA2* stability and metastatic traits with enrichment of cytoplasmic *circNSUN2* in liver metastases relative to primary tumors [179].

An additional circuit relevant to epithelial cancers involves *circITGB6* and IGF2BP3, where TGFβ induction of circITGB6 promotes formation of a complex with IGF2BP3 that stabilizes *PDPN* mRNA and increases PDPN protein, driving EMT features and metastatic dissemination in mouse models [75]. The biochemical plausibility of this stabilization step is supported by the capacity of IGF2BP proteins to bind m^6^A-modified transcripts and enhance their stability and translation under defined conditions [58].

m^6^A also licenses translation of circRNAs in a cap-independent manner, with a single m^6^A site sufficient to initiate translation and with YTHDF3 and eIF4G2 promoting ribosome engagement while METTL3 and METTL14 enhance this output [178]. In colorectal cancer, *circYAP* encodes the YAP-220aa micro-protein in a translation program that requires YTHDF3 and eIF4G2 and that supports invasion and liver metastasis, providing a metastasis-proximal example of m^6^A-dependent circRNA translation [180].

Conversely, m^6^A can designate circRNAs for targeted decay because YTHDF2 together with the adaptor HRSP12 recruits the RNase P or RNase MRP complex to cleave m^6^A-bearing substrates, reducing circRNA abundance and counterbalancing export or translation programs [181].

Hepatocellular carcinoma provides an additional illustration in which *circHPS5* accumulates with increased m^6^A; METTL3 facilitates *circHPS5* production, YTHDC1 supports cytoplasmic export, and *circHPS5* functions as a miRNA sponge that elevates *HMGA2* expression and promotes malignant phenotypes aligned with invasion and colonization [182].

Taken together, YTHDC1-guided export, YTHDF3 and eIF4G2-assisted translation, YTHDF2–HRSP12–RNase P or RNase MRP-mediated decay, and reader-dependent stabilization exemplified by the *circITGB6*–IGF2BP3–*PDPN* and *circNSUN2*–IGF2BP2–*HMGA2* axes outline a coherent framework for how circRNA modifications intersect with metastatic biology; continued work that quantifies per-site modification stoichiometry, reader competition, and in vivo causality is likely to refine these connections across tumor types.

## 5. Conclusions and Future Directions

Metastasis remains the most lethal aspect of cancer, responsible for the majority of deaths in cancer patients. Metastasis is understood to be not only be caused by serial genetic mutations, but also epigenetic plasticity that allows cancer cells to adapt and thrive in new environments. In this review, we highlighted three major epigenetic mechanisms—DNA methylation, RNA modifications, and ncRNAs—and described how their dysregulation drives the multi-step metastatic cascade across various solid tumors. DNA methylation alterations in metastasis can silence key suppressors of invasion while activating prometastatic pathways, and they provide potential biomarkers and targets for intervention. RNA modifications like m^6^A and m^5^C on mRNAs add a post-transcriptional layer of control that cancer cells exploit to fine-tune gene expression for metastasis; targeting the enzymes or readers of these marks is a burgeoning therapeutic concept. ncRNAs, miRNAs, and lncRNAs form extensive regulatory networks that integrate with canonical signaling and epigenetic machinery to promote or restrain metastasis. They not only serve as mechanistic drivers through their effects on gene and protein networks, but also as convenient biomarkers detectable in blood.

A unifying theme is the reversibility and context-dependence of epigenetic modifications. This plasticity is what enables metastatic cells to switch phenotypes like toggling between epithelial and mesenchymal states, and to remain latent or reactivate after years. Importantly, these changes are, in principle, therapeutically reversed.

Epigenetic and RNA-based therapies are still in development, but they offer opportunities for precision medicine approaches to metastasis [183]. For instance, if a patient’s metastasis is found to be driven by overactive m^6^A writer, METTL3, and low miR-200, a combination of a METTL3 inhibitor and a miR-200 mimic could be considered to tackle the disease on two epigenetic fronts. This kind of rationale may soon be testable as our toolkit of epigenetic drugs expands. Looking ahead, continued research is needed to address several challenges and unanswered questions. We must decipher the global epigenomic landscapes of metastases: How do they differ from primary tumors? Single-cell multi-omics and spatial transcriptomics are powerful new approaches to chart heterogeneity within metastases and their surrounding microenvironments. These technologies can reveal how cancer cells at the invasive edge vs. the core of a metastatic lesion differ in chromatin state or ncRNA expression, or how cancer-associated fibroblasts and immune cells in the metastatic niche are epigenetically modulated to support tumor growth. Integrating these layers will provide a holistic view of metastasis and identify critical “epigenetic crosstalk” between tumor cells and stroma.

Another priority is understanding temporal dynamics: which epigenetic changes are early drivers of metastasis, and which are later adaptations or consequences? Discriminating between drivers and passengers enables prioritization of targets. Longitudinal studies can elucidate the sequence of epigenetic events in metastatic progression by analyzing patient samples at the point of initial diagnosis, during remission, and metastatic recurrence.

Epigenetic drugs can have broad effects given that they target fundamental cellular processes. Combining epigenetic therapy with standard treatments (chemotherapy, targeted therapy, immunotherapy) in rational ways will be critical to prevent escape. Epigenetic priming (using a drug to alter tumor cell state to make them more susceptible to another therapy) is an active strategy. Such sequential or combinatorial approaches represent the forefront of translational epigenetics.

Interactions between DNA and RNA modifications and ncRNAs forms an “epigenetic blueprint” that drives metastasis. By encoding this blueprint, we not only gain fundamental insights into how cancer spreads but also open new avenues to intercept the spread of cancer. In recent years, various technologies, such as long-read sequencers and nanopore sequencers, have been developed and have achieved significant results as tools for deciphering these blueprints. While this review does not discuss them in detail, it has become clear that various modifications, such as acetylation and lactylation, influence epigenetic changes, affecting metabolic reprogramming of cancer cells [184,185]. Furthermore, advances in mass spectrometry continue to identify novel RNA modifications. While technological advances are welcome, several caveats temper interpretation. For instance, as antibody-based m^6^A profiling and RNA bisulfite sequencing provide relative enrichment, not absolute stoichiometry, these techniques may misread the sites. A reader-CLIP confirms binding but not causality. Bulk assays confound signals from tumor, stromal, and immune compartments, and many functional studies rely on overexpression or deletion outside physiological ranges. More work using orthogonal chemistries, quantitative standards, and endogenous perturbations is required to build causal maps linking specific marks on defined transcripts or loci to metastatic phenotype in vivo [186].

Epigenetic mechanisms are architects of cellular identity and behavior. However, in cancer, they are hijacked to construct the phenotype of metastasis. Importantly, we are beginning to elucidate these regulatory blueprints, and we can design therapies to rewrite the narrative of metastasis. Multidisciplinary efforts that bridge molecular biology, bioinformatics, and clinical oncology, will be essential to translate epigenetic and ncRNA-targeted interventions from bench to bedside. The ultimate vision is that, in the future, a diagnosis of metastatic cancer will come not with resignation, but with a well-informed, personalized plan to epigenetically disarm the metastases and improve patient outcomes.

## Figures and Tables

**Figure 1 biomolecules-15-01573-f001:**
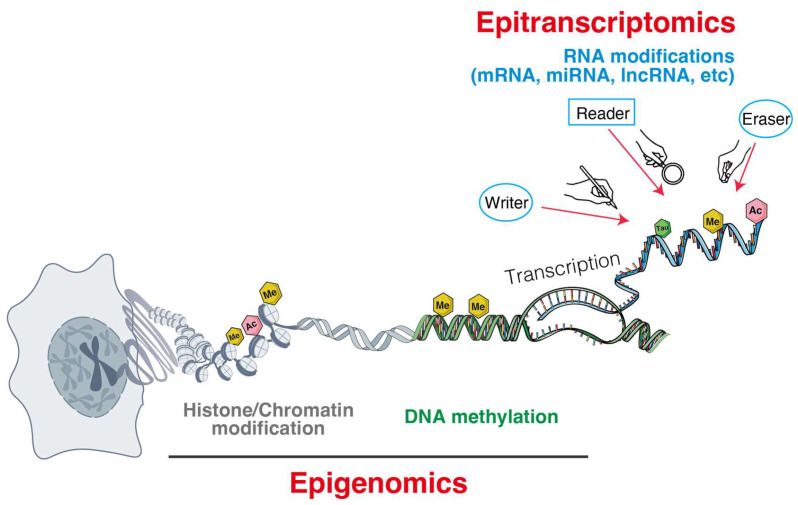
Epigenomic and epitranscriptomic layers orchestrate gene regulation from chromatin to RNA.

**Figure 2 biomolecules-15-01573-f002:**
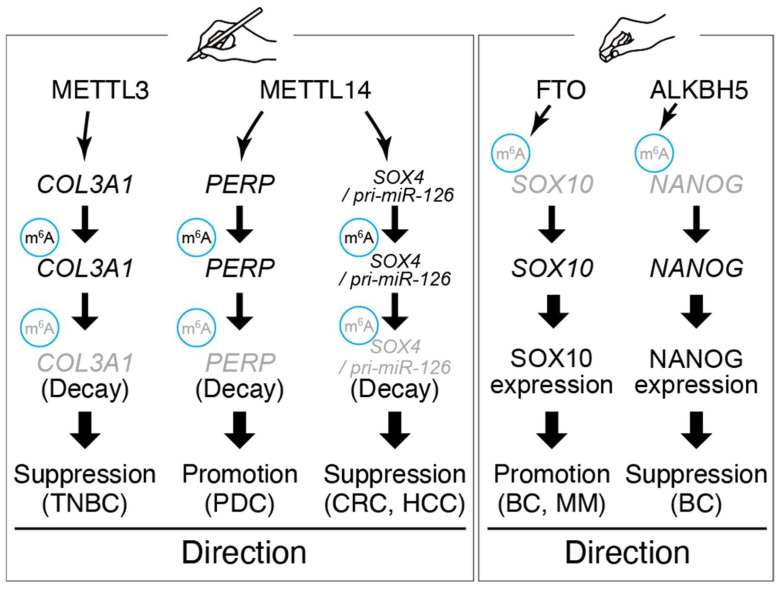
Representative roles and target genes of m^6^A writer and eraser.

**Table 1 biomolecules-15-01573-t001:** DNA methylation-driven modulators of metastasis across cancer types.

Cancer Types	Direction	Target Genes and Molecules	Evidence Type #	References
Breast cancer	Promoting	*CDH1* promoter hypermethylation (silencing)	C, F	[14]
		*DNMT1/3A/3B* overexpression (worse outcome; metastasis risk)	C, F	[15]
Prostate carcinoma	Promoting	*CDH1* promoter hypermethylation (silencing)	C, F	[16]
		*DNMT1/3A/3B* overexpression (high-risk of lymph node metastasis)	C	[17]
Hepatocellular carcinoma	Promoting	DNMT1-mediated *CDH1* promoter hypermethylation (silencing)	F	[18]
		miR-29a-TET–SOCS1–MMP9 axis (Down of 5hmC, Up of MMP9)	C, F, A	[19]
Gastric cancer	Promoting	*TIMP3* promoter methylation	C	[20]
		*CDH1* hypermethylation caused by miR-29b/c-DNMT3A circuit	F, A	[21]
Non-small cell lung cancer	Promoting	*BRMS1* promoter methylation	C	[22]
Pancreatic ductal adenocarcinoma	Promoting	Genome-wide hypomethylation of overexpressed genes	C	[23]
	Suppressive	miR-148a (targets DNMT1; reactivates suppressors)	F	[24,25]
Ovarian carcinoma	Promoting	*DNMT1/3A/3B* overexpression (worse outcome; metastasis risk)	C	[26]

# Evidence type (all that apply): C = clinical association (patient cohorts/tissues); F = functional assays (cell- or tissue-based experiments); A = animal studies (in vivo).

**Table 2 biomolecules-15-01573-t002:** The summary of epitranscriptomic regulators.

Class	Role	Genes
m^6^A	Writers	*METTL3*, *METTL14*, *WTAP*, *VIRMA* (*KIAA1429*), *RBM15*, *RBM15B*
	Erasers	*FTO*, *ALKBH5*
	Readers	*YTHDF1*, *YTHDF2*, *YTHDF3*, *YTHDC1*, *YTHDC2*, *IGF2BP1*, *IGF2BP2*, *IGF2BP3*, *HNRNPA2B1*
m^5^C	Writers	*NSUN2*, *NSUN5*, *NSUN6*, *DNMT2/TRDMT1*
	Erasers	Not established yet
	Readers	*ALYREF*, *YBX1*

**Table 3 biomolecules-15-01573-t003:** m^6^A epitranscriptomics-driven modulators of metastasis across cancer types.

Cancer Types	Direction	Target Genes and Molecules	Evidence Type #	References
Colorectal carcinoma	Promoting	METTL3-mediated m^6^A marks on pri-miR-1246, downregulating SPRED2 to activate MAPK	C, F, A	[64]
		A m^6^A reader IGF2BP3 stabilizes pro-metastatic mRNAs and is a risk marker for poor outcome/metastasis	C	[65]
	Suppressive	METTL14-mediated m^6^A modification on *SOX4* mRNA	C, F, A	[66]
Breast cancer	Promoting	Removal of m^6^A on oncogenic mRNA by FTO to enhance their stability	C, F, A	[67]
		Amplification of m^6^A-enriched pro-metastatic mRNA by YTHDF3	C, F, A	[68]
	Suppressive	Removal of m^6^A on metastasis-suppressive mRNA by ALKBH5	C, F, A	[69]
		Restoring m^6^A modification on *COL3A1* by METTL3, reducing COL3A1 expression	C, F	[70]
Hepatocellular carcinoma	Suppressive	METTL14-mediated m^6^A marks on pri-miR-126 for an effective processing	C, F, A	[71]
Gastric cancer	Promoting	YTHDF1-mediated enhancement of FZD7 translation	C, F, A	[72]
Nasopharyngeal carcinoma	Promoting	Binding of IGF2BP3 to m^6^A-marked *NOTCH3*, stabilizing PDPN	C, F, A	[73,74,75]
Pancreatic ductal adenocarcinoma	Promoting	METTL3-mediated m^6^A modification of *E2F5* mRNA	C, F, A	[76]
		METTL14-mediated m^6^A modification of *PERP* mRNA	C, F, A	[77]
		Overexpression of IGF2BP3, correlated with poor survival	C	[78]
Oral squamous cell carcinoma	Promoting	METTL3-mediated enhancement of BMI1 stability	C, F, A	[79]
Melanoma	Promoting	Demethylation of pro-metastatic mRNA by FTO, stabilizing them	C, F, A	[80]
Prostate cancer	Promoting	RNA decay of tumor suppressive mRNA by YTHDF2, activating AKT signaling	C, F, A	[81]

# Evidence type (all that apply): C = clinical association (patient cohorts/tissues); F = functional assays (cell- or tissue-based experiments); A = animal studies (in vivo).

**Table 4 biomolecules-15-01573-t004:** A m^5^C writer NSUN2–driven modulators of metastasis across cancer types.

Cancer Types	Direction	Target Genes and Molecules	Evidence Type #	References
Bladder cancer	Promoting	NSUN2-mediated m^5^C marks on the *HDGF* 3′-UTR, stabilizing *HDGF* mRNA to enhance growth, invasion, and metastasis	C, F, A	[94,97]
		m^5^C-dependent cross-regulation between NSUN2 and ALYREF, facilitating splicing and stabilization of *RABL6/TK1* mRNA to reinforce carcinoma aggressiveness	C, F, A	[97]
Esophageal squamous cell carcinoma	Promoting	NSUN2-mediated m^5^C stabilization of *GRB2* mRNA in a LIN28B-dependent manner	C, F, A	[98]
		NSUN2-mediated m^5^C on a metastasis-linked lncRNA *NMR*, enabling BPTF association to raise MMPs	C, F	[99,100]
Colorectal carcinoma	Promoting	NSUN2-mediated m^5^C marks on the *SKIL* mRNA, amplifying pro-invasive transcriptional programs	C, F, A	[101]
Prostate cancer	Promoting	A positive epigenetic feedback loop between AR and NSUN2 sustains pro-metastatic signaling and progression.	C, F, A	[102]
Pancreatic ductal adenocarcinoma	Promoting	NSUN2-mediated stability of *TIAM2* mRNA through m^5^C marks, stimulating migratory and invasive programs	C, F, A	[103]
Gastric cancer	Promoting	Stabilization of FOXC2 by NSUN2-associated FOXC2-AS1, enhancing EMT-like program	C, F	[104]
		NSUN2 mediated m^5^C on lncRNA *NR_033928* causes stabilization of *GLS* mRNA, promoting gastric cancer progression	C, F, A	[105]
		NSUN2-mediated m^5^C marks on the *ORAI2* mRNA by a peritoneal high-fat diet, fostering peritoneal dissemination	C, F, A	[106]

# Evidence type (all that apply): C = clinical association (patient cohorts/tissues); F = functional assays (cell- or tissue-based experiments); A = animal studies (in vivo).

**Table 5 biomolecules-15-01573-t005:** Key factors in miRNA functions.

Class	Role	Genes
miRNA	Biogenesis	*DROSHA*, *DGCR8*, *DICER1*
	Effector	*AGO2*, *LIN28A/B*
	Modulators	*ZCCHC11* (*TUT4*), *ZCCHC6* (*TUT7*)), *ADAR, ADARB1*

**Table 6 biomolecules-15-01573-t006:** Representative lncRNAs that remodel the epigenome and epitranscriptome in metastasis.

lncRNA	Direction	Mechanism	Cancer Types	References
*HOTAIR*	Promoting	HOTAIR scaffolds PRC2 and LSD1/CoREST to raise H3K27me3 and reduce H3K4 methylation at anti-metastatic loci, reprogramming chromatin toward invasion and EMT.	Breast cancer, Colorectal carcinoma, Gastric cancer, Hepatocellular carcinoma	[148,149,150]
*GClnc1*	Promoting	Scaffolding of recruiting WDR5 (H3K4me3) and KAT2A/GCN5 (acetylation) at target promoters, boosting EMT/migration/invasion	Gastric cancer	[151]
*GCAWKR*	Promoting	Scaffolding of chromatin-modification factors to activate oncogenic transcription programs that support progression.	Gastric cancer	[152]
*SChLAP1*	Promoting	Antagonizes SWI/SNF occupancy genome-wide, shifting chromatin toward pro-metastatic transcriptional states; high expression tracks aggressive disease.	Prostate cancer	[153]
*NEAT1*	Promoting	m^6^A and ALKBH5-mediated demethylation increase NEAT1 activity to modulate EZH2/EMT programs; m^6^A on *NEAT1* also enhances bone colonization in prostate models.	Gastric cancer, Prostate cancer	[154,155,156,157,158]
*KCNK15-AS1*	Suppressive	ALKBH5-dependent demethylation stabilizes metastasis-restraining *KCNK15-AS1*, curbing migration/invasion via PTEN–AKT tuning.	Pancreatic ductal adenocarcinoma	[159,160]
*MIR100HG* (miR-100 host)	Promoting	Binding of HNRNPA2B1 to stabilize m^6^A-modified *TCF7L2* mRNA, amplifying Wnt signaling and invasion; disrupting METTL3-installed m^6^A or this interaction suppresses malignancy.	Colorectal carcinoma	[131,161,162,163]
*XIST*	Context- dependent	m^6^A promotes XIST-mediated transcriptional repression; in CRC #, XIST drives EMT/metastasis, while METTL14 can suppress by down-regulating oncogenic XIST.	Colorectal carcinoma	[82,164,165]
*NKILA*	Context- dependent	In cholangiocarcinoma, m^5^C/m^6^A-modified *NKILA* is stabilized by YBX1 and promotes migration via the miR-582-3p–YAP1 axis; in NSCLC #, NKILA antagonizes NF-κB/SNAIL to inhibit migration/invasion.	Cholangiocarcinoma, NSCLC, Breast cancer, Tongue/oral squamous cell carcinoma	[121,122,166]
*FOXC2-AS1*	Promoting	Associates with NSUN2 to stabilize *FOXC2* mRNA, enhancing EMT-like programs and invasiveness.	Gastric cancer	[104]

# CRC; Colorectal Cancer, NSCLC; Non-Small Cell Lung Cancer.

## Data Availability

Not applicable.

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
