# Peer review of "Rewriting the Fate of Cancer: Epigenetic and Epitranscriptomic Regulators in the Metastatic Cascade"

_biomolecules, 2025, doi:10.3390/biom15111573_

Round 1

Reviewer 1 Report

Comments and Suggestions for Authors

This review explains how cancer spreads in the body (metastasis) by focusing not only on gene mutations but also on “switches” that turn genes on or off without changing DNA itself—such as chemical tags on DNA, modifications on RNA, and noncoding RNAs that control other genes. These changes are flexible and sometimes reversible, making them powerful drivers of cancer’s ability to invade, survive, and grow in new places. The authors highlight how studying these processes can reveal new ways to detect, predict, and treat cancer spreads. The topic is interesting, however, there are still several comments that need to be addressed before considering publication.

1.The authors claims that metastasis accounts for “90% of cancer deaths” (pp1–2, lines 31–41). However, reference [3] (Dillekås et al., Cancer Medicine 2019) explicitly questions the validity of this 90% figure. Could the authors clarify how this reference supports the statement?

2.The review does not describe databases, search terms, time frame, or inclusion/exclusion criteria; even for a narrative review, a minimal level of search transparency is important. Could the authors clarify this aspect?

3.Although the title, introduction, and keywords all emphasize circRNAs, the manuscript provides only a passing mention of a single IGF2BP3–circITGB6 case and lacks any dedicated subsection or systematic review. Could the authors clarify this imbalance?

4.The manuscript devotes disproportionate attention to m⁶A (METTL3/IGF2BPs) and m⁵C (NSUN2–ALYREF/YBX1) modifications, while other important epitranscriptomic marks—such as pseudouridine (Ψ), m¹A, ac⁴C, m⁷G, and mRNA-level A-to-I editing—are scarcely discussed, which leaves the epitranscriptomic landscape incomplete. Could the authors clarify whether these marks were deliberately excluded or considered outside the scope?

5.Classical epigenetic mechanisms (histone modifications, chromatin remodeling) are mentioned only sporadically, mostly through lncRNA-related cases, and lack a systematic synthesis. Would the authors consider providing a more structured overview of these mechanisms?

6.In Table 3, the content and references are not consistently aligned. For example, the entry “ALKBH5 demethylation … suppresses metastasis” under breast cancer is linked to refs. [72], [78], which actually describe METTL3–COL3A1 and FTO–BNIP3 rather than ALKBH5. In addition, the table classifies METTL14 as pro-metastatic in pancreatic cancer, whereas the text describes it as “rarely reported to promote metastasis.” Could the authors carefully verify that each table entry is supported by the correct citation and that the directionality is consistent with the narrative?

7.The manuscript frequently uses strong expressions such as “promising drug targets” or “able to suppress dissemination,” yet the level of supporting evidence is not clearly defined. Could the authors clarify the evidence tier to avoid overinterpretation?

8.Terminology is inconsistent (gene/protein format, ncRNA naming, modification abbreviations). Could the authors unify usage?

9.Tables 1 and 4 currently list genes and regulators but do not convey the strength of supporting evidence. Adding columns for “evidence type (clinical association / functional assays / animal studies), sample size, model, effect direction, and evidence level” would allow readers to more readily assess the robustness of each finding. Could the authors consider this addition?

Author Response

Comment #1-1

The authors claims that metastasis accounts for “90% of cancer deaths” (pp1–2, lines 31–41). However, reference [3] (Dillekås et al., Cancer Medicine 2019) explicitly questions the validity of this 90% figure. Could the authors clarify how this reference supports the statement?

Response #1-1

We are grateful for this insightful comment. It has long been believed that metastasis accounts for over 90% of cancer deaths. This claim is indeed asserted in representative papers such as References 1 and 2. However, in recent years, papers like Reference 3 have raised questions. That said, the study was limited to Norway and lacked sufficient investigative reporting data, so it does not overturn existing reports. Considering this situation, while we cite Reference 3. However, we removed the phrase "90% of cancer deaths", aiming to align the manuscript and references with current trends in cancer research.

(Page 1, lines 31)

Comment #1-2

The review does not describe databases, search terms, time frame, or inclusion/exclusion criteria; even for a narrative review, a minimal level of search transparency is important. Could the authors clarify this aspect?

Response #1-2

We appreciate you for your comment. We added a section at the end of the revised manuscript that outlines the databases, search terms, time frame, and inclusion/exclusion keywords to demonstrate transparency in the search process for this manuscript.

(Page 22, lines 752-783)

Comment #1-3

Although the title, introduction, and keywords all emphasize circRNAs, the manuscript provides only a passing mention of a single IGF2BP3–circITGB6 case and lacks any dedicated subsection or systematic review. Could the authors clarify this imbalance?

Response #1-3

We apologize for the insufficient description of circular RNAs (circRNAs) in the original manuscript and agree with your suggestion. We added a section on circular RNAs (circRNAs) to the revised manuscript, describing the importance of circRNAs and their modifications.

(Page 19-20, lines 620-665)

Comment #1-4

The manuscript devotes disproportionate attention to m⁶A (METTL3/IGF2BPs) and m⁵C (NSUN2–ALYREF/YBX1) modifications, while other important epitranscriptomic marks—such as pseudouridine (Ψ), m¹A, ac⁴C, m⁷G, and mRNA-level A-to-I editing—are scarcely discussed, which leaves the epitranscriptomic landscape incomplete. Could the authors clarify whether these marks were deliberately excluded or considered outside the scope?

Response #1-4

Epitranscriptome markers include over 170 distinct types, such as pseudouridine (Ψ), m¹A, ac⁴C, m⁷G, and A-to-I editing at the mRNA level. However, comprehensively covering all of them in this review is difficult. Therefore, we focused this review specifically on the most abundant RNA modifications, m6A and m5C. We added a note to this reason in the revised manuscript.

(Page 11-12, lines 181-182, and 184-185)

Comment #1-5

Classical epigenetic mechanisms (histone modifications, chromatin remodeling) are mentioned only sporadically, mostly through lncRNA-related cases, and lack a systematic synthesis. Would the authors consider providing a more structured overview of these mechanisms?

Response #1-5

We appreciate your thoughtful point that histone modifications and chromatin remodeling warrant a systematic synthesis. We fully agree that these foundational mechanisms are central to cancer biology. However, our review was intentionally scoped to address a more focused question: how epigenetic and epitranscriptomic regulation, with emphasis on RNA modifications and their crosstalk with noncoding RNAs, rewires each step of the metastatic cascade. A broad, stand-alone survey of classical pathways would substantially expand the manuscript and, in our judgment, dilute the central contribution of this work, which is a mechanism-first, metastasis-centric integration of DNA/RNA modification circuitry with stage mapping. As shown in the following, there are excellent, comprehensive reviews that already synthesize histone and chromatin remodeling biology at length, and we prefer to direct readers to those authoritative sources rather than reproduce a general overview here. In light of this scope and space rationale, we have not added a new systematic section on classical epigenetics. We acknowledge that the reviewer’s suggestion is entirely reasonable, and we are grateful for it. We hope the editor and reviewers will view our decision as a deliberate choice to preserve clarity and focus on the metastasis-specific intersections that constitute the manuscript’s main value.

  • Kouzarides T. Chromatin modifications and their function. Cell. 2007;128(4):693-705.
  • Clapier CR, Iwasa J, Cairns BR, Peterson CL. Mechanisms of action and regulation of ATP-dependent chromatin-remodelling complexes. Nat Rev Mol Cell Biol. 2017;18(7):407-422.
  • Dawson MA, Kouzarides T. Cancer epigenetics: from mechanism to therapy. Cell. 2012;150(1):12-27.

Comment #1-6

In Table 3, the content and references are not consistently aligned. For example, the entry “ALKBH5 demethylation … suppresses metastasis” under breast cancer is linked to refs. [72], [78], which actually describe METTL3–COL3A1 and FTO–BNIP3 rather than ALKBH5. In addition, the table classifies METTL14 as pro-metastatic in pancreatic cancer, whereas the text describes it as “rarely reported to promote metastasis.” Could the authors carefully verify that each table entry is supported by the correct citation and that the directionality is consistent with the narrative?

Response #1-6

We apologize for the citation error in Table 3. It appears that the reference numbers shifted during the paper revision process. We carefully checked all references in the Tables 1, 3, 4, and 6. Also, we revised that the table entry matches the content of the revised manuscript.

(Page 3, Table 1; Page 7-8, Table 3; Page 9, lines 246-248; Page 10, Table 4; Page 16-17, Table 6)

Comment #1-7

The manuscript frequently uses strong expressions such as “promising drug targets” or “able to suppress dissemination,” yet the level of supporting evidence is not clearly defined. Could the authors clarify the evidence tier to avoid overinterpretation?

Response #1-7

We thank you for pointing out the expressions in the manuscript. We have made revisions to entries based on evidence (Page 3, Lines 160-167) and to sections where it was difficult to determine their appropriateness (Page 1, lines 20-21; Page 13, lines 413; Page 20, lines 690).

Comment #1-8

Terminology is inconsistent (gene/protein format, ncRNA naming, modification abbreviations). Could the authors unify usage?

Response #1-8

We apologize for the inconsistent usage of terminology. We carefully revised the terminology of the gene/protein format, ncRNA name, and modification abbreviations.

(All revised sections are highlighted within the revised manuscript.)

Comment #1-9

Tables 1 and 4 currently list genes and regulators but do not convey the strength of supporting evidence. Adding columns for “evidence type (clinical association / functional assays / animal studies), sample size, model, effect direction, and evidence level” would allow readers to more readily assess the robustness of each finding. Could the authors consider this addition?

Response #1-9

We appreciate your considerable comment and suggestion. We agreed with your suggestion. We added the new columns for "evidence type" in Tables 1, 3, and 4. Due to column and chart size limitations, only the evidence type was added, but the strength of the supporting evidence was emphasized.

(Page 3, Table 1; Page 7-8, Table 3; Page 10, Table 4)

Note:
The revised manuscript was proofread by Dr. Anisimov Sergei to improve the English.

Reviewer 2 Report

Comments and Suggestions for Authors

Summary:

This is a nicely written review that draws attention to the growing body of evidence that the progress of cancer and specifically metastasis is very dependent on epigenetic changes. Importantly, it considers a wide range of molecular pathways, so it starts with classical DNA-centered epigenetic processes but then goes into a lot of detail on RNA modifications and the role of regulatory noncoding RNAs. So I believe that it is a very useful contribution to the field because it rightly draws attention to these complexities, as some colleagues may be less familiar with the "RNA biology" aspects of epigenetics in cancer.

Major points:

  1. The review is very well researched and highly interesting, but perhaps a slight criticism is that in covering a lot, some of it comes across as a bit of a list, and this means that sometimes specific examples are not covered sufficiently. Could you consider trimming down some sections, and providing more mechanistic detail, eg see minor point 5. Some paragraphs are also extremely long.
  2. On a related point, there are some nice tables, but what is missing is some diagrams / figures that might really add to the review by presenting a visual representation of the processes covered.

Minor points:

  1. Figure 1, the red titles, "epitranscriptomics" and "epigenomics" are a bit too prominent, could the font size be reduced and the figure rebalanced. Also "RNA modifications" is a bit simplistic, could there be more detail here in terms of the key molecular processes referred to.
  2. Line 106, hypomethylation activating some key genes - can you provide specific examples.
  3. Line 160, please elaborate on what you mean by precise targeting, perhaps illustrating with an example.
  4. Table 2 appears a quite simple list of genes, without detail in terms of the nature of the writers / erasers and I wonder if it could be enhanced.
  5. Line 220 onwards - this paragraph is extremely long, and quite a thick list of facts. Could it be restructured slightly to aid clarity. The problem is when you cover so much, sometimes some points are not explained sufficiently, for example, Line 2060 "IGF2BP3 was found to partner with a circRNA (circITGB6) to stabilize PDPN mRNA, leading to heightened EMT and 261 metastasis" - very interesting but how does it do that?
  6. Line 398 can you add a further specification, technically all RNAs that aren't translated are "noncoding" so you need to differentiate here between "housekeeping" ncRNAs (eg rRNA, snoRNA, tRNA etc) and "regulatory" ncRNAs (eg miRNAs, lncRNAs etc).
  7. In section 3 you mention circRNAs at the outset but then don't really cover them much and I wonder if you could add more information on them, and how they could fit the whole narrative.
  8. I'm not sure why you would need a separate "Discussion" section, which I would expect to see in a research paper instead of a review. It is also oddly short, and then there is a much longer conclusions and future directions section. What I think might work is if you had a more specific and detailed section on the therapeutic implications, and then a more brief and separate future work and conclusion section at the end.

Author Response

Major points:

Comment #2-1

The review is very well researched and highly interesting, but perhaps a slight criticism is that in covering a lot, some of it comes across as a bit of a list, and this means that sometimes specific examples are not covered sufficiently. Could you consider trimming down some sections, and providing more mechanistic detail, eg see minor point 5. Some paragraphs are also extremely long.

Response #2-1

We appreciate your considerable suggestion. We revised the manuscript to enhance reader comprehension by adding new figures and organizing content through section breaks. Additionally, we added specific details to particular descriptions.

(Page 2, Figure 1; Page 8, Figure 2; Page 19, lines 622-667)

Comment #2-2

On a related point, there are some nice tables, but what is missing is some diagrams / figures that might really add to the review by presenting a visual representation of the processes covered.

Response #2-2

We appreciate your thoughtful comment. We added a new Figure (Figure 2) and modified Figure 1 for a better understanding of the content in the manuscript.

(Page 2, Figure 1; Page 8, Figure 2)

Minor points:

Comment #2-3

Figure 1, the red titles, "epitranscriptomics" and "epigenomics" are a bit too prominent, could the font size be reduced and the figure rebalanced. Also "RNA modifications" is a bit simplistic, could there be more detail here in terms of the key molecular processes referred to.

Response #2-3

We are grateful for your helpful suggestion. We modified Figure 1 by changing the font size and adding the key roles of the modulators.

(Page 2, Figure 1)

Comment #2-4

Line 106, hypomethylation activating some key genes - can you provide specific examples.

Response #2-4

We are thankful for your detailed suggestion. We added the specific genes activated by hypomethylation to the revised manuscript.

(Page 4, lines 108-110)

Comment #2-5

Line 160, please elaborate on what you mean by precise targeting, perhaps illustrating with an example.

Response #2-5

We appreciate the important suggestion. To facilitate understanding of what precise targeting means here, we have provided details using specific examples.

(Page 5, lines 160-167)

Comment #2-6

Table 2 appears a quite simple list of genes, without detail in terms of the nature of the writers / erasers and I wonder if it could be enhanced.

Response #2-6

We appreciate your valuable suggestion. To make the content in Table 2 more intuitive, we have revised Figure 1 and added a new Figure 2.

(Page 2, Figure 1; Page 8, Figure 2)

Comment #2-7

Line 220 onwards - this paragraph is extremely long, and quite a thick list of facts. Could it be restructured slightly to aid clarity. The problem is when you cover so much, sometimes some points are not explained sufficiently, for example, Line 2060 "IGF2BP3 was found to partner with a circRNA (circITGB6) to stabilize PDPN mRNA, leading to heightened EMT and 261 metastasis" - very interesting but how does it do that?

Response #2-7

We are grateful for the constructive suggestion. To help readers better understand this session, we have added Figure 2. We have also revised the paragraph to make it simpler. Also, we moved the circRNA topic from this paragraph to a new section.

(Page 8, lines 236; Page 19-20, lines 620-665)

Comment #2-8

Line 398 can you add a further specification, technically all RNAs that aren't translated are "noncoding" so you need to differentiate here between "housekeeping" ncRNAs (eg rRNA, snoRNA, tRNA etc) and "regulatory" ncRNAs (eg miRNAs, lncRNAs etc).

Response #2-8

We appreciate this astute suggestion. We added a specification of noncoding RNAs to the revised manuscript.

(Page 13, lines 420-422)

Comment #2-9

In section 3 you mention circRNAs at the outset but then don't really cover them much and I wonder if you could add more information on them, and how they could fit the whole narrative.

Response #2-9

We appreciate you for raising the important point. We added a section on circular RNAs (circRNAs) to the revised manuscript, describing their importance and modifications.

(Page 19-20, lines 620-665)

Comment #2-10

I'm not sure why you would need a separate "Discussion" section, which I would expect to see in a research paper instead of a review. It is also oddly short, and then there is a much longer conclusions and future directions section. What I think might work is if you had a more specific and detailed section on the therapeutic implications, and then a more brief and separate future work and conclusion section at the end.

Response #2-10

We sincerely appreciate the constructive suggestion. We combined the Discussion and Conclusions and Future Directions sections, removing the Discussion section. We believe that the therapeutic significance of epigenetics and RNA modifications is still evolving. Therefore, in the section on conclusions and future directions, we have enriched the content by including technical developments, issues, and points that could not be discussed in this review.

(Page 21, lines 723-738)

Reviewer 3 Report

Comments and Suggestions for Authors

Dear Author,

I would like to congratulate you on your review article entitled “Rewriting the Fate of Cancer: Epigenetic and Epitranscriptomic Regulators in the Metastatic Cascade.” It is a thorough, well-structured, and comprehensive piece of work. However, I recommend a few minor revisions to enhance the manuscript’s clarity and impact.

Minor Comment:

  • Please reduce the number of references, as several of them contain overlapping information (e.g., References 1, 2, 4, and 6). Including multiple references with similar content is not necessary.

Major Comments:

  • Recent advances in long-read RNA/DNA sequencing technologies (such as Oxford Nanopore) have significantly reshaped the field of cancer epigenetics. The authors are encouraged to include major discoveries from these studies to strengthen the review.
  • The manuscript primarily focuses on methylation-related epigenetic changes but overlooks other important modifications such as acetylation (Ref 2) and lactylation (Ref 3). Incorporating recent findings in these areas would make the review more comprehensive and engaging for readers.

Best regards,

Sujan

Regerence:

  1. Lau BT, Almeda A, Schauer M, et al. Single-molecule methylation profiles of cell-free DNA in cancer with nanopore sequencing. Genome Med. 2023;15(1):33. Published 2023 May 3. doi:10.1186/s13073-023-01178-3
  2. Wu Z, Guan KL. Acetyl-CoA, protein acetylation, and liver cancer. Mol Cell. 2022;82(22):4196-4198. doi:10.1016/j.molcel.2022.10.015
  3. Chen J, Huang Z, Chen Y, et al. Lactate and lactylation in cancer. Signal Transduct Target Ther. 2025;10(1):38. Published 2025 Feb 12. doi:10.1038/s41392-024-02082-x

Author Response

Major points

Comment #3-1

Recent advances in long-read RNA/DNA sequencing technologies (such as Oxford Nanopore) have significantly reshaped the field of cancer epigenetics. The authors are encouraged to include major discoveries from these studies to strengthen the review.

Response #3-1

We appreciate your thoughtful suggestion. Recent technological advances have significantly transformed the field of cancer epigenetics. We believe it is important to discuss the technical aspects to detail these discoveries. To avoid overcomplicating the discussion in this review, we have briefly outlined these developments and challenges in the section on Conclusions and Future Directions.

(Page 21, lines 723-726 and 729-738)

Comment #3-2

The manuscript primarily focuses on methylation-related epigenetic changes but overlooks other important modifications such as acetylation (Ref 2) and lactylation (Ref 3). Incorporating recent findings in these areas would make the review more comprehensive and engaging for readers.

Response #3-2

We totally agree with your point. Our review intentionally scope to address a more focused question: how epigenetic and epitranscriptomic regulation, with emphasis on RNA modifications and their crosstalk with noncoding RNAs, rewires each step of the metastatic cascade. On the other hand, since acetylation and lactylation are important topics, we briefly described them in the section on Conclusions and Future Directions, citing appropriate references.

(Page 21, lines 726-729)

Minor point

Comment #3-3

Please reduce the number of references, as several of them contain overlapping information (e.g., References 1, 2, 4, and 6). Including multiple references with similar content is not necessary.

Response #3-3

We are grateful to the reviewer for this helpful suggestion. We removed as many of the multiple references with similar content as possible. The number of reference papers has not changed significantly from the original manuscript, due to the addition of sections and revisions to the manuscript.

Round 2

Reviewer 1 Report

Comments and Suggestions for Authors

Thanks for the authors’ responses. Most of the previous comments have been successfully addressed, and the revised manuscript has been remarkably improved. However, there are still several remaining issues that need to be further corrected.

1.In Table 3, the role of ALKBH5 in BC is seems incorrectly listed as promoting, whereas both Figure 2 and the main text clearly describe ALKBH5 as suppressing metastasis.

2.In Figure 2, METTL14 is labeled as “promotion (PDC, HCC)”, but both the text and Table 3 indicate that METTL14 promotes metastasis in PDC yet suppresses it in HCC. Please check.

3.In the abbreviation list, 5hmC and m5C are defined as “5-hydroxymethylcytidine” and “5-methylcytidine,” respectively, which correspond to RNA nucleosides. However, in the DNA methylation section, these symbols clearly refer to the DNA base forms (5-hydroxymethylcytosine and 5-methylcytosine). Please check.

4.In the “Literature search and selection” section, the authors limited their PubMed queries with the filter review[pt], which would only retrieve review articles. However, the manuscript primarily relies on original research studies. Please revise.

Author Response

Reviewer #1

Comment #1-1

In Table 3, the role of ALKBH5 in BC is seems incorrectly listed as promoting, whereas both Figure 2 and the main text clearly describe ALKBH5 as suppressing metastasis.

Response #1-1

Thank you for pointing out our mistake. During revision, the Table 3 entry was not updated appropriately to match the finalized text and figure. We have corrected Table 3 by relocating ALKBH5 (breast cancer) under “Suppressive” and verified consistency across the text and Figure 2.

(Page 7, Table 3)

Comment #1-2

In Figure 2, METTL14 is labeled as “promotion (PDC, HCC)”, but both the text and Table 3 indicate that METTL14 promotes metastasis in PDC yet suppresses it in HCC. Please check.

Response #1-2

We appreciate your careful reading of our revised manuscript. We have revised Figure 2 to align with Table 3 and the accompanying text, thereby enhancing clarity and understanding.

(Page 8, Figure 2)

Comment #1-3

In the abbreviation list, 5hmC and m5C are defined as “5-hydroxymethylcytidine” and “5-methylcytidine,” respectively, which correspond to RNA nucleosides. However, in the DNA methylation section, these symbols clearly refer to the DNA base forms (5-hydroxymethylcytosine and 5-methylcytosine). Please check.

Response #1-3

Our abbreviation list initially followed an RNA-centric convention (m5C/hm5C for nucleosides), while the DNA section used the base conventions (5mC/5hmC). Due to an error during the bulk conversion step in the first revision, the symbols of DNA and RNA modifications were mixed. We have standardized the nomenclature and corrected all occurrences throughout the text.

(Page 23, lines 793-797)

Comment #1-4                                                                               

In the “Literature search and selection” section, the authors limited their PubMed queries with the filter review[pt], which would only retrieve review articles. However, the manuscript primarily relies on original research studies. Please revise.

Response #1-4

We agree and have revised the “Literature search and selection” subsection to describe a two-stage approach. Our initial scoping searches used the review[pt] filter to map the field. Then we performed primary study searches without that filter and built the tables from original research. We have revised “Literature search and selection” accordingly and now explicitly separate Stage 1 (scoping reviews) from Stage 2 (primary-study search).

(Page 22, lines 755-757)

Reviewer 2 Report

Comments and Suggestions for Authors

Dear authors, many thanks for addressing all of my points comprehensively, it is much appreciated. 

Author Response

Comment #2-1

Dear authors, many thanks for addressing all of my points comprehensively, it is much appreciated.

Response #2-1

Thank you very much for your valuable comments. We appreciate the time and energy you spent.